# Post-Training Quantization for Vision Transformer

**Zhenhua Liu**[1,2], **Yunhe Wang**[2]*, **Kai Han**[2], **Wei Zhang**[2], **Siwei Ma**[1,3], **Wen Gao**[1,3]

[1]School of Electronic Engineering and Computer Science, Peking University
[2] Huawei Noah's Ark Lab [3]Peng Cheng Laboratory
liu-zh@pku.edu.cn, {yunhe.wang, kai.han, wz.zhang}@huawei.com,
{swma, wgao}@pku.edu.cn

## Abstract

Recently, transformer has achieved remarkable performance on a variety of computer vision applications. Compared with mainstream convolutional neural networks, vision transformers are often of sophisticated architectures for extracting powerful feature representations, which are more difficult to be developed on mobile devices. In this paper, we present an effective post-training quantization algorithm for reducing the memory storage and computational costs of vision transformers. Basically, the quantization task can be regarded as finding the optimal low-bit quantization intervals for weights and inputs, respectively. To preserve the functionality of the attention mechanism, we introduce a ranking loss into the conventional quantization objective that aims to keep the relative order of the self-attention results after quantization. Moreover, we thoroughly analyze the relationship between quantization loss of different layers and the feature diversity, and explore a mixed-precision quantization scheme by exploiting the nuclear norm of each attention map and output feature. The effectiveness of the proposed method is verified on several benchmark models and datasets, which outperforms the state-of-the-art post-training quantization algorithms. For instance, we can obtain an 81.29% top-1 accuracy using DeiT-B model on ImageNet dataset with about 8-bit quantization. Code will be available at https://gitee.com/mindspore/models/tree/master/research/cv/VT-PTQ.

## 1 Introduction

Following the applications in Natural Language Processing (NLP) tasks, transformer-based models have shown great power in various Computer Vision (CV) tasks, such as image classification [11, 26], object detection [4, 39] and image super-resolution [5]. Pre-trained with large-scale data, these models usually have hundreds of millions of parameters. For instance, there are 307M parameters and 64G FLOPs in the ViT-L model, which is both memory and computation expensive during inference. This brings great challenges for these models to run on resource-constrained devices like mobile phones and intelligent cars. Besides, the real-time computer vision applications that integrate transformer-based models have to meet low latency requirements to achieve a high quality customer experience. Therefore, the model compression technology of transformer-based models is urgently needed for deployment in industrial environments.

Among various compression methods like pruning [16, 29, 19] and weight decomposition [37], quantization method [9, 38, 7, 27, 14, 32] compresses a neural network by using lower bit-width for weight values without changing the model architecture, which is particularly useful for carefully-designed network architectures like transformers. Quantizing both weights and inputs can speed up inference by tuning floating-point operations into integer or bit operations. There have been some

---

*Corresponding author

35th Conference on Neural Information Processing Systems (NeurIPS 2021).

training-aware quantization approaches for transformer-based models in NLP (*e.g.*, BERT [17]) [34, 23, 35, 22]. However, these methods are not designed for computer vision tasks and usually need additional training or fine-tuning. Furthermore, in some scenarios, the entire training data is not available to optimize the quantization model and the training costs for edge devices are intolerable.

Post-training quantization [24] is a kind of efficient model compression technique, which can directly quantize neural network models without fine-tuning. Most of the existing post-training quantization methods are designed for convolutional neural networks [3, 21, 30] or recurrent neural networks [36]. These methods do not take the character of vision transformer into consideration (*e.g.*, the attention mechanism do not exist in CNNs), which are not perfectly suitable for quantizing vision transformer. However, vision transformers are showing stronger performance in a large variety of computer vision tasks. Thus, we are motivated to explore the post-training quantization for them to reduce the costs on memory and computation.

In this paper, we study the post-training quantization method for vision transformer models with mixed-precision for higher compression and speed-up ratios. The quantized process in the transformer is formulated as an optimization problem for finding the optimal quantization intervals. Specially, our goal is to maximize the similarity between the full-precision and quantized outputs in vision transformers. To better preserve the functionality of the attention mechanism, we thoroughly analyze the difference between attention layers and conventional layers such as MLP. Then, a ranking loss is introduced to keep the relative order of attention values. Furthermore, we propose to determine the bit-widths of each layer according to the feature diversity, *i,e,*, the nuclear norm calculated by the attention map and output features. We alternatively search the quantization intervals of weights and inputs in all layers to obtain the best quantization results. In addition, bias correction is introduced to diminish the cumulative quantization error. Experimental results on several benchmarks demonstrate the effectiveness of our algorithm for achieving better performance over the state-of-art post-training quantization approaches.

## 2 Related Works

Here, we reviews the transformer-based models designed for computer vision tasks. And the training-aware quantization schemes proposed for BERT and post-training quantization algorithms are summarized and analyzed.

### 2.1 Vision Transformer

Inspired by the major success of transformer architectures in the field of NLP, researchers have recently applied transformer to computer vision (CV) tasks [13]. Chen *et al.* [6] trained a sequence transformer to auto-regressively predict pixels, achieving results comparable to CNNs on image classification tasks. Another vision transformer model is ViT, which applies a pure transformer directly to treat image patches as the sequences. Recently proposed by Dosovitskiy *et al.* [11], it has achieved great performance on multiple image recognition benchmarks. Touvron *et al.* [26] produce competitive convolution-free transformers by training on ImageNet only while introducing a teacher-student strategy specific to transformers. In addition to basic image classification, transformer has been utilized to address a variety of other computer vision problems, including object detection [4, 39], semantic segmentation [5], image processing [5], and video understanding [5]. Han et al. [15] proposed a Transformer-iN-Transformer (TNT) model for modeling both patch-level and pixel-level representation. Tang *et al.* [25] proposed an augmented shortcut scheme to improve the performance of vision transformers. Thanks to its exceptional performance, more and more researchers are proposing transformer-based models for a wide range of computer vision tasks.

### 2.2 Compression of Transformer in NLP

Owing to the remarkable performance of BERT in many NLP tasks, many researchers have tried to compress the model to reduce the memory and computation complexity of BERT. Wu et al. [31] proposed Short Range Attention (LSRA) to conduct transformer on edge devices, where one group of heads specializes in the local context modeling (by convolution) while another group specializes in the long-distance relationship modeling. In [22, 34], 8-bit quantization is successfully applied to Transformer-based models with comparable performance as the full-precision baseline. However,

quantizing these models to ultra low bits (*e.g.*, 1 or 2 bits) can be much more challenging due to significant reduction in model capacity. To avoid severe accuracy drop, more complex quantization methods, like mixed-precision quantization [23, 33] and product quantization (PQ) [12] are used. In addition, Zhang *et al.* [35] propose TernaryBERT, which use both approximation-based and loss-aware ternarization methods and empirically investigate the ternarization granularity of different parts of BERT. Moreover, to reduce the accuracy degradation, they also leverage the knowledge distillation technique. Bai *et al.* [1] further push BERT quantization to the limit with weight binarization. They propose ternary weight splitting, which initializes the binary model by equivalent splitting from a half-sized ternary network. However, these methods are not designed for computer vision tasks and need additional training or fine-tuning.

### 2.3 Post-Training Quantization

There are many works focusing on developing post-training quantization methods, without any training or fine-tuning. In particular, Yoni *et al.* [8] propose the OMSE method to optimize the $L_2$ distance between the quantized tensor and the original tensor. Moreover, Ron *et al.* [2] present the so-called ACIQ method to analytically compute the clipping range, as well as the per-channel bit allocation for NNs. Zhao *et al.* [36] propose an outlier channel splitting (OCS) method to solve the outlier channel problem. Wang *et al.* [28] propose a Bit-Split and Stitching framework for lower-bit post-training quantization and an Error Compensated Activation Quantization method, which could lower the quantization error for activations. Nagel *et al.* [20] propose AdaRound, a weight-rounding mechanism for post-training quantization that adapts to the data and the task loss. By approximating the task loss with a Taylor series expansion, the rounding task is posed as a quadratic unconstrained binary optimization problem. The recent work of [21] propose Data-Free Quantization, which further pushes post-training quantization to zero-shot scenarios, where neither training nor testing data are accessible during quantization. Cai *et al.* [3] introduce ZeroQ, which distills an input data distribution to match the statistics in the batch normalization layers of the model and utilize a Pareto Frontier method to select automatically the bit-precision configuration of mixed-precision settings. These methods are designed for CNNs and do not consider the unique structure of vision transformers such as self-attention layers.

## 3 Methodology

In this section, we elaborate on the proposed mixed-precision post-training quantization scheme for the vision transformer. The similarity-aware quantization for linear layers and ranking-aware quantization for self-attention layers are presented. In addition, the bias correction method for optimization and the mixed-precision quantization based on nuclear norm of the attention map and output feature are introduced.

### 3.1 Preliminaries

A standard transformer receives an input as a 1-D sequence of token embeddings, so the vision transformers usually reshape the image $\mathbf{I} \in \mathbb{R}^{H \times W \times C}$ into a sequence of flatted 2D patches $I^p \in \mathbb{R}^{n \times (P^2 \cdot C)}$. Here, $H$ and $W$ are the height and width of the original image and $(P, P)$ is the resolution of each image patch, $n = \frac{HW}{P^2}$ is then the effective sequence length for the transformer. Usually, the vision transformers use constant widths through all of its layers, so a trainable linear projection maps each vectorized patch to the model dimension $d$. Thus, the input to the first transformer layer is:

$$\mathbf{X}_1 = [x_{class}; I_1^p \mathbf{W}_1^E; \cdots; I_n^p \mathbf{W}_n^E] + \mathbf{E}^{pos}. \tag{1}$$

$$\text{where } \mathbf{W}^E \in \mathbb{R}^{(P^2 \cdot C) \times d}, \mathbf{E}^{pos} \in \mathbb{R}^{(n+1) \times d} \tag{2}$$

A standard transformer layer includes two main modules: Multi-Head Self Attention (MSA) and Multi-Layer Perceptron (MLP) module. For the $l$-th transformer layer, suppose the input to it is $\mathbf{X}_l \in \mathbb{R}^{n \times d}$, the attention scores computed by the dot product of queries and keys can be formulated as:

$$\mathbf{A}_l = \mathbf{Q}_l \mathbf{K}_l^{\mathrm{T}} = \mathbf{X}_l \mathbf{W}_l^Q \mathbf{W}_l^{K^{\mathrm{T}}} \mathbf{X}_l^{\mathrm{T}}. \tag{3}$$

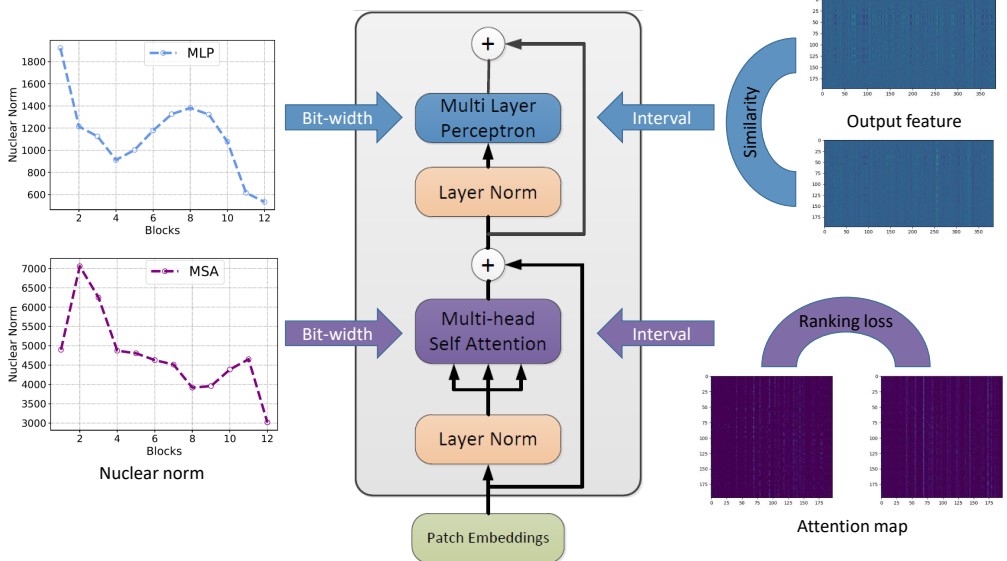

Figure 1: Diagram of the proposed mixed-precision post-training quantization method for vision transformer. The similarity-aware and ranking-aware quantization are designed for finding the optimal quantization interval of the linear operations and self-attention layers. The bit-widths of transformer layers are determined based on the nuclear norm of the attention map and the output feature.

Then the softmax function is applied on the normalized scores to get the output and the output of the multi-head self attention module is:

$$\text{MSA}(\mathbf{X}_l) = \text{Softmax}(\frac{1}{\sqrt{d}}\mathbf{A}_l)\mathbf{X}_l\mathbf{W}_l^V \cdot \mathbf{W}_l^O. \tag{4}$$

The MLP module contains two linear layers parameterized by $\mathbf{W}^1 \in \mathbb{R}^{d \times d_f}, b^1 \in \mathbb{R}^{d_f}$ and $\mathbf{W}^2 \in \mathbb{R}^{d_f \times d}, b^2 \in \mathbb{R}^d$ respectively, where $d_f$ is the number of neurons in the intermediate layer of MLP. Denote the input to MLP as $\mathbf{Z}_l \in \mathbb{R}^{n \times d}$, the output is then computed as:

$$\text{MLP}(\mathbf{Z}_l) = \text{GeLU}(\mathbf{Z}_l\mathbf{W}^1 + b^1)\mathbf{W}^2 + b^2. \tag{5}$$

Combining Eq. (4) and (5), the forward propagation for the $l$-th transformer layer can be formulated as:

$$\mathbf{Z}_l = \mathbf{X}_l + \text{MSA}(\text{LN}(\mathbf{X}_l)), \tag{6}$$
$$\mathbf{X}_{l+1} = \mathbf{Z}_l + \text{MLP}(\text{LN}(\mathbf{Z}_l)), \tag{7}$$

where LN represents the layer normalization.

The most computational costs of vision transformer lie on the large matrix multiplication in MSA and MLP module. Following the mainstream quantization methods for CNNs [7, 21], we quantize all the weights and inputs involved in matrix multiplication. For weight quantization, we quantize the weights $\mathbf{W}^Q, \mathbf{W}^K, \mathbf{W}^V, \mathbf{W}^O, \mathbf{W}^1, \mathbf{W}^2$ in Eq. (4) and (5) for all transformer layers, as well as the linear embedding $\mathbf{W}^E$ in Eq. (1). Besides these weights, we also quantize the inputs of all linear layers and matrix multiplication operations. Following the methods in [22, 35], we do not quantize the softmax operation and layer normalization, because the parameters contained in these operations are negligible and quantizing them may bring significant accuracy degradation.

## 3.2 Ranking-Aware Post-Training Quantization

For post-training quantization, we need to restrict the floating-numbers to a finite set of values. The choice of quantization intervals is critical for quantization and one popular option is to use a uniform quantization function, where the data range is equally split:

$$\Psi_\Delta(\mathbf{Y}) = \text{Clamp}(\text{Round}(\frac{\mathbf{Y}}{\Delta}), -2^{b-1}, 2^{b-1} - 1), \tag{8}$$

where $\Delta$ is the quantization interval, $b$ is the quantization bit-width and $\mathbf{Y}$ is a tensor representing weights or inputs. Clamp denotes that elements in the tensor that exceed the ranges of the quantized domain are clipped.

For the layers in vision transformer, the original output can be computed as $\mathbf{O}_l = \mathbf{X}_l \mathbf{W}_l$. The uniform quantization for the weights and inputs and the corresponding dequant operation can be described as:

$$\widehat{\mathbf{O}}_l = \Psi_{\Delta_l^X}(\mathbf{X}_l)\Psi_{\Delta_l^W}(\mathbf{W}_l) \cdot \Delta_l^W \cdot \Delta_l^X, \tag{9}$$

where $\widehat{\mathbf{O}}_l$ denotes the outputs of the quantized layer. From Eq. (8) and Eq. (9), it can be seen that the quantization intervals actually control the clamping thresholds in quantization process, which affects the quantization results to a great extent. Therefore, we are motivated to focus on optimizing the quantization intervals for both weights $\Delta_l^W$ and inputs $\Delta_l^X$, where inputs $X_l$ are generated from a given calibration dataset $\mathbf{D}$ with $N$ samples. Specifically, the calibration dataset is much less than the common training dataset.

The self-attention layer is the critical component of the transformer since it can calculate the global relevance of the features, which makes the transformer unique from the convolutional neural networks. For the calculation of self-attention (Eq. 3), we empirically find that the relative order of the attention map has been changed after quantization as shown in Fig 1, which could cause a significant performance degradation. Thus, a ranking loss is introduced to solve this problem during the quantization process:

$$\mathcal{L}_{ranking} = \sum_{k=1}^{h} \sum_{p=1}^{w-1} \sum_{q=p+1}^{w} \Phi((\widehat{\mathbf{A}}_{kp} - \widehat{\mathbf{A}}_{kq}) \cdot sign(\mathbf{A}_{kp} - \mathbf{A}_{kq})), \tag{10}$$

in which $\Phi(m) = (\theta - m)_+$ is hinge function with parameter $\theta$, $(h, w)$ are the size of matrix $\mathbf{A}$. Given a pair of examples, the loss is $0$ only when the examples are in the correct order and differed by a margin.

Then we combine the ranking loss with the similarity-aware quantization, and the overall optimization goal can be described as:

$$\min_{\Delta_l^W, \Delta_l^X} \gamma \cdot \mathcal{L}_{ranking} - \frac{1}{N} \sum_{i=1}^{N} \Gamma(\mathbf{O}_l^i, \widehat{\mathbf{O}}_l^i), \quad s.t. \ \Delta_l^W, \Delta_l^X \in \mathbb{R}^+ \tag{11}$$

where $\mathcal{L}_{rank}$ denote the pairwise ranking based loss function, and $\gamma$ is the trade-off hyper-parameter. $\Gamma(\mathbf{O}_l^i, \widehat{\mathbf{O}}_l^i)$ denotes the similarity metric between the original and quantized output feature maps, which can be formulated as:

$$\Gamma(\widehat{\mathbf{O}}, \mathbf{O}) = \frac{\sum_{j=1}(\mathbf{O}_j - \overline{\mathbf{O}})(\widehat{\mathbf{O}}_j - \overline{\widehat{\mathbf{O}}})}{\sqrt{\sum_{j=1}(\mathbf{O}_j - \overline{\mathbf{O}})^2}\sqrt{\sum_{j=1}(\widehat{\mathbf{O}}_j - \overline{\widehat{\mathbf{O}}})^2}}, \tag{12}$$

where the Pearson correlation coefficient is adopted as the measurement for the similarity since it subtracts the mean value of the data and can be more representative for the similarity between the distribution of quantized and original feature maps.

To solve the above optimization problem, we present a simple but efficient alternative searching method for the uniform quantization of transformer layers. Firstly, the quantization interval of inputs $\Delta_l^X$ is fixed, and the quantization interval of weights $\Delta_l^W$ is optimized for adjustment. Secondly, $\Delta_l^W$ is fixed, and $\Delta_l^X$ is optimized to fine-tune the quantization interval of the inputs. $\Delta_l^W$ and $\Delta_l^X$ are alternately optimized until the target function converges or the maximum iteration is exceeded. Moreover, for fast convergence, $\Delta_l^W$ and $\Delta_l^X$ are initialized in terms of the maximum of weights or inputs respectively. For the search space of $\Delta_l^W$ and $\Delta_l^X$, we linearly divide interval of $[\alpha\Delta_l, \beta\Delta_l]$ into $C$ candidate options and conduct a simple search strategy on them.

**Bias Correction** To further reduce the biased error for the outputs raised by quantization, a bias correction method is then introduced after each search iteration. Suppose the quantization error of weights and inputs are defined as:

$$\epsilon^X = \Psi_{\Delta^X}(\mathbf{X}) \cdot \Delta^X - \mathbf{X}, \tag{13}$$

$$\epsilon^W = \Psi_{\Delta^W}(\mathbf{W}) \cdot \Delta^W - \mathbf{W}. \tag{14}$$

If the expectation of the error for output is not zero, then the mean of the output will change. This shift in distribution may lead to detrimental behavior in the following layers. We can correct this change by seeing that:

$$\mathbb{E}[\widehat{\mathbf{O}}] = \mathbb{E}[\mathbf{O}] + \mathbb{E}[\epsilon^W \mathbf{X}] + \mathbb{E}[\epsilon^X \mathbf{W}] + \mathbb{E}[\epsilon^X \epsilon^W]. \tag{15}$$

Thus, subtracting the expected error on the output from the biased output ensures that the mean for each output unit is preserved. For implementation, the expected error can be computed using the calibration data and subtracted from the layer's bias parameter, since the expected error vector has the same shape as the layer's output.

### 3.3 Nuclear Norm Based Mixed-Precision Quantization

Different transformer layers are attending to different structures, and it is expected that they exhibit different sensitivity. Thus, assigning the same number of bit-widths to all the layers is sub-optimal. As a result, we explore mixed-precision quantization, where more bits are assigned to more sensitive layers in order to retain performance. Considering the unique structure of transformer layer, we assign all the operations in the MSA or MLP modules with the same bit-width. This will also be friendly to the hardware implementation since the weights and inputs are assigned with the same bit-width.

Singular value decomposition (SVD) is an important matrix decomposition approach in linear algebra. It takes a rectangular matrix of gene expression data, whose formulation can be written as :

$$\mathbf{M} = \mathbf{U}\boldsymbol{\Sigma}\mathbf{V}, \quad \text{tr}(\mathbf{M}) = \sum_{i=1}^{m} \boldsymbol{\Sigma}_{ii}, \tag{16}$$

where the diagonal entries $\boldsymbol{\Sigma}_{ii}$ of $\boldsymbol{\Sigma}$ are known as the singular values of $\mathbf{M}$. And the nuclear norm $\mathbf{tr}$ is the sum of singular values, which represents the data relevance of the matrix. In this paper, we propose to estimate the sensitivity of the transformer layer with the nuclear norm of the attention map in the MSA module and the output feature in the MLP module. The nuclear norm can be used to reduce the search space of the mixed-precision settings, while using higher bit-widths for layers that are more sensitive and vice versa. Inspired by the method in [10], we utilize a Pareto frontier approach to determine the bit-width. The main idea is to sort each candidate bit-width configuration based on the total second-order perturbation that they cause, according to the following metric:

$$\Omega = \sum_{i=1}^{L} \Omega_i = \sum_{i=1}^{L_{MHA}} \text{tr}(\mathbf{A}_i) \cdot \|\widehat{\mathbf{A}_i} - \mathbf{A}_i\|_2^2 + \sum_{j=1}^{L_{MSA}} \text{tr}(\mathbf{O}_j) \cdot \|\widehat{\mathbf{O}_j} - \mathbf{O}_j\|_2^2. \tag{17}$$

Given a target model size, the candidate bit-width configurations are sorted based on their $\Omega$ value and choose the bit-width configuration with minimal $\Omega$. The nuclear norm of the attention map and output feature in each transformer layer are shown in Figure 1. As we can see, they are various for different transformer layers.

## 4 Exprimental results

In this section, we evaluate the performance of the proposed post-training quantization scheme on vision transformer model for image classification (ViT [11] and DeiT [26]) and object detection (DETR [4]). To the best of our knowledge, there is no published work done on post-training quantization of vision transformer at this point, so we implement recent post-training quantization methods for CNNs as described in the papers by ourselves. It is shown that the proposed method outperforms the conventional post-training quantization methods. Moreover, extensive experiments of ablation study have shown that the proposed similarity-aware, ranking-aware quantization and bias correction method are beneficial for the post-training quantization of vision transformer.

### 4.1 Implementation details

**Datasets** For image classification, the CIFAR-10, CIFAR-100 and ILSVRC-2012 ImageNet (we refer to it as ImageNet in what follows) datasets are utilized to evaluate the quantization performance.

The CIFAR-10 dataset consists of $50K$ training images and $10K$ test images, which are labeled for 10 classes. And CIFAR-100 dataset also contains $50K$ training images and $10K$ test images, expect that they are labeled for 100 classes. ImageNet dataset contains 1.2 million training images and $50K$ validation images labeled for 1,000 categories. For object detection task, the COCO2017 dataset is utilized to evaluate the quantization performance, which contains $118K$ training images and $5K$ validation images.

**Experimental settings** We randomly select 100 images for CIFAR-10 and CIFAR-100 dataset and 1000 images for ImageNet and COCO2017 dataset from the training dataset as the calibration dataset. For the hyper-parameter, $\alpha$ and $\beta$ are set to 0.5 and 1.2 for all the experiments. The maximum iteration is set to 20 if not mentioned specifically. For mixed-precision, we utilize {4,5,6,7,8} and {6,7,8,9,10} bits while the target bit-width are 6 bit and 8 bit, respectively.

**Baseline** For image classification, we evaluate our quantization method on two popular vision transformer implementation: ViT [11] and DeiT [26]. The ViT-B, ViT-L, DeiT-S, DeiT-B are adopted as the baseline model, whose top-1 accuracy on ImageNet dataset are 71.58%, 71.48%, 79.8%, 81.8% respectively. For a fair comparison, we utilize the official implementation of DeiT and do not use other techniques like knowledge distillation. For object detection, the DETR model using ResNet-50 backbone is adopted, which achieves a 42.0 mAP on COCO dataset.

## 4.2 Results and Analysis

**Image classification** The experimental results are shown in Table 1. We firstly evaluate the proposed method on ViT-B and ViT-L model. ViT-B model is a 12-layer transformer with 12 heads and 768 embedding dimension. For the similar quantized model size, the proposed method outperforms percentile-based method [18] by 3.35% and 2.07% on CIFAR-10 dataset, respectively. And it is worth noting that the performance of the proposed 8-bit model is comparable to the full-precision model. The proposed method obtains the similar performance on CIFAR-100 dataset and ImageNet dataset, while the average gains are 2.95% and 3.28% respectively. Moreover, the performance of the proposed 6-bit model is even better than the 8-bit percentile-based model, which means that the proposed method can save about 25% memory and 44% computational costs than conventional post-training quantization method.

ViT-L model is much larger network which consists of 24 transformer layer with 16 heads and 1024 embedding dimension. It contains 307M parameters, however its performance is worse than ViT-B. We also test the quantization methods on CIFAR-10, CIFAR-100 and ImageNet dataset. As shown in Table 1, the performance of the proposed method outperforms the percentile-based method by a large margin. It is worth mentioning that the 8-bit proposed model is even better than full-precision model on CIFAR-10 dataset and comparable to the full-precision model on CIFAR-100 dataset and ImageNet model. It is supposed that there is more redundancy in the ViT-L model and the performance degradation of quantization is less than that of ViT-B model.

The architecture of DeiT network is the same as ViT, expect that DeiT utilizes the data augmentation and regularization strategies. As a result, the performance of DeiT is much better than ViT. Among the models, ViT-S consists of 12 transformer layers with 6 heads and 384 embedding dimension. As we can see, the percentile-based method largely hurts the performance while the accuracy losses of 6-bit and 8-bit models are 9.31% and 5.82%. EasyQuant [30] is a popular simple post-training quantization method which improves the performance loss to 6.54% and 3.21%, respectively. Bit-Split proposes a bit splitting and stitching framework [28], while the Top-1 accuracy degradation are 5.76% and 2.74%. In comparison, the Top-1 accuracy losses of the proposed post-training quantization scheme are 5.22% and 2.33% respectively. In addition, when the mixed-precision is conducted, the 8-bit quantized model can achieve 78.09% Top-1 accuracy.

DeiT-B is a much larger network than DeiT-S, which consists of 12 transformer layers with 12 heads and 768 embedding dimension. As shown in Table 1, the Top-1 accuracy of percentile-based are 73.99% and 75.21% when quantized to 6-bit and 8-bit respectively. And the proposed scheme improves the performance of the quantized model to 77.47% and 81.29%. Another point is that the accuracy losses of DeiT-B are smaller than DeiT-S and we think that this is because DeiT-B consists of more parameters and is more representative when quantized to the same bit-width.

Table 1: Comparison on the performance of proposed mixed-precision post-training quantization method with conventional quantization method for image classification. 'MP' represents for mixed-precision.

| Model | Dataset | Method | W-bit | A-bit | Model size (MB) | Top-1 Accuracy |
|---|---|---|---|---|---|---|
| ViT-B | CIFAR-10 | Baseline | 32 | 32 | 344 | 98.13 |
| | | Percentile | 6 | 6 | 64.5 | 93.48 |
| | | Ours | 6 MP | 6 MP | 64.6 | **96.83** |
| | | Percentile | 8 | 8 | 86.2 | 95.72 |
| | | Ours | 8 MP | 8 MP | 86.0 | **97.79** |
| | CIFAR-100 | Baseline | 32 | 32 | 344 | 87.13 |
| | | Percentile | 6 | 6 | 64.5 | 80.56 |
| | | Ours | 6 MP | 6 MP | 64.4 | **83.99** |
| | | Percentile | 8 | 8 | 86.2 | 83.28 |
| | | Ours | 8 MP | 8 MP | 86.5 | **85.76** |
| | ImageNet | Baseline | 32 | 32 | 344 | 77.91 |
| | | Percentile | 6 | 6 | 64.5 | 71.58 |
| | | Ours | 6 MP | 6 MP | 64.8 | **75.26** |
| | | Percentile | 8 | 8 | 86.2 | 74.10 |
| | | Ours | 8 MP | 8 MP | 86.5 | **76.98** |
| ViT-L | CIFAR-10 | Baseline | 32 | 32 | 1228 | 97.86 |
| | | Percentile | 6 | 6 | 230.2 | 93.27 |
| | | Ours | 6 MP | 6 MP | 232 | **96.09** |
| | | Percentile | 8 | 8 | 307 | 94.19 |
| | | Ours | 8 MP | 8 MP | 305.8 | **97.90** |
| | CIFAR-100 | Baseline | 32 | 32 | 1228 | 86.35 |
| | | Percentile | 6 | 6 | 230.2 | 80.54 |
| | | Ours | 6 MP | 6 MP | 231 | **83.69** |
| | | Percentile | 8 | 8 | 307 | 83.01 |
| | | Ours | 8 MP | 8 MP | 307.8 | **85.83** |
| | ImageNet | Baseline | 32 | 32 | 1228 | 76.53 |
| | | Percentile | 6 | 6 | 230.2 | 71.48 |
| | | Ours | 6 MP | 6 MP | 231.6 | **75.46** |
| | | Percentile | 8 | 8 | 307 | 75.17 |
| | | Ours | 8 MP | 8 MP | 306.4 | **76.41** |
| DeiT-S | ImageNet | Baseline | 32 | 32 | 88 | 79.8 |
| | | Percentile [18] | 6 | 6 | 16.5 | 70.49 |
| | | EasyQuant [30] | 6 | 6 | 16.5 | 73.26 |
| | | Bit-Split [28] | 6 | 6 | 16.5 | 74.04 |
| | | Ours | 6 | 6 | 16.5 | **74.58** |
| | | Ours | 6 MP | 6 MP | 16.6 | **75.10** |
| | | Percentile [18] | 8 | 8 | 22.0 | 73.98 |
| | | EasyQuant [30] | 8 | 8 | 22.0 | 76.59 |
| | | Bit-Split [28] | 8 | 8 | 22.0 | 77.06 |
| | | Ours | 8 | 8 | 22.0 | **77.47** |
| | | Ours | 8 MP | 8 MP | 22.2 | **78.09** |
| DeiT-B | ImageNet | Baseline | 32 | 32 | 344 | 81.8 |
| | | Percentile [18] | 6 | 6 | 64.5 | 73.99 |
| | | EasyQuant [30] | 6 | 6 | 64.5 | 75.86 |
| | | Bit-Split [28] | 6 | 6 | 64.5 | 76.39 |
| | | Ours | 4 MP | 4 MP | 43.6 | 75.94 |
| | | Ours | 6 | 6 | 64.5 | **77.02** |
| | | Ours | 6 MP | 6 MP | 64.3 | **77.47** |
| | | Percentile [18] | 8 | 8 | 86.0 | 75.21 |
| | | EasyQuant [30] | 8 | 8 | 86.0 | 79.36 |
| | | Bit-Split [28] | 8 | 8 | 86.0 | 79.42 |
| | | Ours | 8 | 8 | 86.0 | **80.48** |
| | | Ours | 8 MP | 8 MP | 86.8 | **81.29** |

Table 2: Comparison on the performance of proposed mixed-precision post-training quantization method with conventional quantization method for DETR. 'MP' represents for mixed-precision.

| Model | Dataset | Method | W-bit | A-bit | Model size (MB) | mAP |
|-------|---------|--------|-------|-------|-----------------|-----|
| | | Baseline | 32 | 32 | 164 | 42.0 |
| | | Percentile [18] | 6 | 6 | 30.75 | 37.5 |
| | | EasyQuant [30] | 6 | 6 | 30.75 | 39.0 |
| | | Bit-Split [28] | 6 | 6 | 30.75 | 38.9 |
| | | Ours | 6 | 6 | 30.75 | **40.1** |
| DETR | COCO2017 | Ours | 6 MP | 6 MP | 30.98 | **40.5** |
| | | Percentile [18] | 8 | 8 | 41.00 | 38.6 |
| | | EasyQuant [30] | 8 | 8 | 41.00 | 40.4 |
| | | Bit-Split [28] | 8 | 8 | 41.00 | 40.6 |
| | | Ours | 8 | 8 | 41.00 | **41.2** |
| | | Ours | 8 MP | 8 MP | 41.64 | **41.7** |

**Object Detection** In order to show the generalization capability of proposed method, we also evaluate our method for object detection task using DETR [4]. The experimental results are shown in Table 2. As we can see, the proposed method outperforms percentile-based method, EasyQuant, Bit-Split by 2.6, 1.1 and 1.2 mAP for 6-bit quantization, respectively. The mixed-precision quantization can further boost the performance of the method. For 8-bit quantization, the mAP of the proposed mixed-precision quantization method is comparable to the full-precision model.

## 4.3 Ablation study

In this section, we evaluate the effect of the proposed similarity-aware quantization module, ranking-aware quantization module, bias correction method and the mixed-precision method. The experimental results are shown in Table 3, while experiments are conducted on ImageNet dataset with ViT-B model. As we can see, the Top-1 accuracy of only using similarity-aware quantization is 75.42% which is inferior to the full-precision model and using ranking-aware quantization loss and bias correction method can improve the performance by 0.52% and 0.39%. It is worth noting that the nuclear norm based mixed-precision can further promote the performance of the quantized model, since it considers the variant sensitivity of different layers.

It is also shown that the Top-1 accuracy of using the similarity-aware mixed-precision quantization is 76.26%. And the ranking-aware quantization and bias correction can still boost the performance in this case. Besides, the performance of the 8-bit quantized model using all the proposed methods is 76.98%, which is comparable to the full-precision model.

Table 3: Ablation study of the proposed similarity-aware quantization module, ranking-aware quantization module, bias correction and mixed-precision method.

| Model | Similarity | Ranking | Bias Correction | Mixed-Precision | Model size (MB) | Top-1 Accuracy |
|-------|-----------|---------|-----------------|-----------------|-----------------|----------------|
| | – | – | – | – | 344 | 77.91 |
| | ✓ | ✗ | ✗ | ✗ | 86.2 | 75.42 |
| | ✓ | ✓ | ✗ | ✗ | 86.2 | 75.94 |
| | ✓ | ✗ | ✓ | ✗ | 86.2 | 75.81 |
| ViT-B | ✓ | ✓ | ✓ | ✗ | 86.2 | 76.49 |
| | ✓ | ✗ | ✗ | ✓ | 86.5 | 76.26 |
| | ✓ | ✗ | ✓ | ✓ | 86.5 | 76.61 |
| | ✓ | ✓ | ✗ | ✓ | 86.5 | 76.53 |
| | ✓ | ✓ | ✓ | ✓ | 86.5 | **76.98** |

We also compared the performance of the proposed method with the hessian-based mixed-precision method, where the experiments are conducted with ViT-B on ImageNet dataset. As we can see in Table 4, the proposed method achieves a similar result while the consuming computation time is much less than hessian-based approach.

Table 4: Performance comparison with hessian-based mixed=precision method.

| Method | W-bit | A-bit | Model size (MB) | Computation time (s) | Top-1 Accuracy |
|---|---|---|---|---|---|
| Hessian-based | 8 MP | 8 MP | 86.7 | 754.6 | 77.01 |
| Ours | 8 MP | 8 MP | 86.5 | 53.1 | 76.98 |

## 5 Conclusion

In this paper, we have developed a novel post-training quantization scheme for vision transformer, in which the bit-widths of each layer are variant based on the nuclear norm of the attention map and output feature in the transformer layer. To solve the optimization problem of the quantization, we propose to search the optimal quantization interval for remaining the similarity between the quantized and original feature maps. In addition, we thoroughly analyze the different between attention layers and conventional layers and introduce a ranking loss to keep the relative order of the attention values. Specifically, the bias correction is employed to reduce the accumulated quantization error. Last but not the least, the optimal quantization interval for each transformer layer is carefully optimized using an alternative searching strategy. Experimental results show that the proposed method outperforms the conventional post-training quantization method by a large margin in terms of both network accuracy and memory costs.

## Acknowledge

This work was partly supported by the National Natural Science Foundation of China (61961130392) and PKU-Baidu Fund(2019BD003). Besides, High-Performance Computing Platform of Peking University is acknowledged.

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
