# OpenReview forum: "Post-Training Quantization for Vision Transformer"
_NeurIPS.cc/2021/Conference — NeurIPS 2021 Poster_

### Official Review · Reviewer_YhTZ · 2021-07-14

**Rating:** 7
**Confidence:** 1

**Summary:**

The authors propose a novel quantization algorithm for vision transformer by searching for the best quantization intervals of inputs and weights. The optimization is conducted by minimizing the similarity loss and the ranking loss. The bias correction method is employed and the nuclear norm based mixed-precision is proposed. The proposed method is evaluated on the image classification and object detection tasks.

**Limitations And Societal Impact:**

I have one question that the proposed method can be directly used on CNN except the proposed ranking-aware loss. Have you compared the method on CNN and does it have the same beneficial?
Another question is what’s the meaning of Y in Eq.(18)? Is it the attention map or the output feature?


**Main Review:**

Generally, this is a novel and effective quantization method for vision transformer, since the training and inference costs of vision transformer are vast. The proposed ranking-aware loss is specially designed for vision transformer and the mixed-precision quantization is instructive. The experimental results of the proposed quantization method outperforms the post-training quantization methods of CNN.

**Time Spent Reviewing:**

5

---

> ### Author Response · Authors · 2021-08-10
> **Authors' Response**
>
> Thanks for the positive and helpful comments.
>
> $\textbf{Q1:}$ Is the proposed method can be directly used on CNN except the proposed ranking-aware loss. Have you compared the method on CNN and does it have the same beneficial?
>
> $\textbf{A1:}$ Thanks for the nice question. We didn’t conduct the experiments on the CNN and the proposed method can be conducted on the CNN theoretically. However, we want to demonstrate the effectiveness of proposed method on vision transformer in this paper.
>
> $\textbf{Q2:}$ What’s the meaning of Y in Eq.(18)? Is it the attention map or the output feature?
>
> $\textbf{A2:}$ Thanks for the question. The Y in Eq.(18) represents the output feature for MLP module and the attention map for the MSA module.

---

### Official Review · Reviewer_bDSX · 2021-07-14

**Rating:** 6
**Confidence:** 4

**Summary:**

This paper proposes a post-training quantization scheme for vision transformer, which determines the bit-width of each layer according to the nuclear norm of the attention map and output feature. In order to find the optimal quantization interval, the authors develops two loss to optimize the quantization errors.


**Limitations And Societal Impact:**

(1)	What are the optimization goal of the self-attention layer? Are they the weights and inputs of the Q and K? There are some details need to be provided in Eq.(12).
(2)	Does the attention map in Eq.(13) the same as the output in Eq.(12)?
(3)	Some related works need to be discussed more concretely, such as quantization method for the transformer [1] [2].
[1] Fully Quantized Transformer for Improved Translation.
[2] Efficient 8-bit quantization of transformer neural machine language translation model.


**Main Review:**

(1)	The similarity-aware loss and bias correction are reasonable and the designed ranking-aware loss designed for the self-attention layer is novel.
(2) The authors propose a mixed-precision quantization method for the MSA and MLP modules and experimental results demonstrate the effectiveness of the proposed method
(3) This paper is well organized.


**Time Spent Reviewing:**

7

---

> ### Author Response · Authors · 2021-08-10
> **Authors' Response**
>
> We would like to thank for the encouraging and constructive comments. Here are responses to the questions.
>
> $\textbf{Q1:}$ What are the optimization goal of the self-attention layer? Are they the weights and inputs of the Q and K? There are some details need to be provided in Eq.(12).
>
> $\textbf{A1:}$ Thanks for the nice question. The optimization goal of the self-attention layer are the quantization scale of the weight and activation of Q and K. We also use the alterative searching method except that the four variables need to optimized.
>
> $\textbf{Q2:}$ Does the attention map in Eq.(13) the same as the output in Eq.(12)?
>
> $\textbf{A2:}$ Sorry for the misleading. They both represent the attention map in the MSA module and we use the output O in Eq.(12) to keep the same description as in Eq.(10).
>
> $\textbf{Q3:}$ Some related works need to be discussed more concretely, such as quantization method for the transformer [1] [2].
>
> $\textbf{A3:}$ Thanks for the suggestion. [1] aimed to exploit hardware resources as efficiently as possible, quantizing all operations which could provide a computational speed gain. [2] proposed to replace 32-bit floating point computations with 8-bit integers (INT8) by transforming the FP32 computational graph and present a parallel batching technique to maximize CPU utilization during inference. However, these quantization methods in NLP are training-aware, while we propose a post-training quantization for vision transformer. We will discuss the works in the manuscript.
>
> [1] Fully Quantized Transformer for Improved Translation.
>
> [2] Efficient 8-bit quantization of transformer neural machine language translation model.

---

> > ### Comment · Reviewer_bDSX · 2021-08-28
> > **I will keep my score.**
> >
> > Thanks for the response. I think my concerns have been addressed.

---

### Official Review · Reviewer_oxRc · 2021-07-15

**Rating:** 7
**Confidence:** 5

**Summary:**

The authors have proposed a post-training quantization method for vision transformer, where different modules employs the different bit-width. The quantization intervals are determined by optimizing the similarity between the quantized and original output features. In addition, the ranking-aware loss is proposed to promote the quantization performance of the self-attention layers.

**Limitations And Societal Impact:**

The authors adequately addressed the limitations and potential negative societal impact of their work.

**Main Review:**

Strengths:
The paper is well organized and I think it has significant contribution to the field. The proposed similarity and ranking-aware loss are rational. The experiments are well evaluated and they have demonstrated the effectiveness of the proposed method.

Weaknesses:
- What is the optimization process of the mixed-precision? I have a little confusion about the Eq.(18) and what is the meaning of the Y in Eq.(18)?
- What is the configuration of the mixed-precision in the experiments since you have used 6 MP and 8 MP?

**Time Spent Reviewing:**

5

---

> ### Author Response · Authors · 2021-08-10
> **Authors' Response**
>
> We would like to thank for the encouraging feedbacks.
>
> $\textbf{Q1:}$ What is the optimization process of the mixed-precision? I have a little confusion about the Eq.(18) and what is the meaning of the Y in Eq.(18)?
>
> $\textbf{A1:}$ Thanks for the nice question. There are lots of candidate bit-width configurations in the search space, so we utilize a Pareto Frontier approach to find the bit-width configuration with the minimal $\Omega$ as described in the following equation (Eq.(18) in the paper).
>
> $\Omega = \sum_{i=1}^L \Omega_i=\sum_{i=1}^L \sum_{j=1}^{m}\sigma_j(\textbf{Y})\cdot\|\widehat{\textbf{Y}}-\textbf{Y}\|^2_2.$
>
> where L is the number of layers and m is the number of singular values.
>
> The Y in Eq.(18) represents the output feature of MLP module and attention map for the MSA module.
>
> $\textbf{Q2:}$ What is the configuration of the mixed-precision in the experiments since you have used 6 MP and 8 MP?
>
> $\textbf{A2:}$ We manually set the candidate bit-widths to {4,5,6,7,8} and {6,7,8,9,10} for 6 MP and 8 MP, which are commonly used in other methods.

---

### Official Review · Reviewer_NVLj · 2021-07-16

**Rating:** 3
**Confidence:** 5

**Summary:**

This paper proposed a post-training quantization algorithm for vision transformers. The algorithm introduces a ranking loss to improve the quantization performance of the model by keeping the relative order of self-attention results. Besides, this paper explores a mixed-precision quantization scheme by exploiting the nuclear norm of each attention map and output feature.  The proposed method is verified on several benchmark models and datasets.

**Limitations And Societal Impact:**

No. I did not find clear discussions on the limitations and potential negative societal impact of the work.

**Main Review:**

Pros:
- The paper proposes a post-training quantization method for vision transformer and applies it to different CV tasks.
- Ablation studies on ranking loss, bias correction and mixed-precision quantization are welcome.
- The paper is clear to read.

Cons:
- The authors adopt Pearson correlation coefficient as the measurement for the similarity (L156-162). There are many other simpler choices for computing similarity (cosine, Euclidean, KL, etc). For example, the work of [22] uses cosine similarity [22], and the previous work of "Improving Post Training Neural Quantization: Layer-wise Calibration and Integer Programming" uses Euclidean distance. Why do the authors choose the Pearson correlation coefficient, and what are the advantages of this metric?
- Regarding ranking-aware quantization for self-attention, the authors state that "we empirically find that the relative order of the attention map has been changed after quantization as shown in Fig 1, which could cause a significant performance degradation" (L166-167). This "observation" is crucial to the main contribution and novelty of this paper, but it lacks theoretical analysis and sufficient experimental evaluations to demonstrate that the change of order actually negatively affects the final quantization accuracy. Without such analysis or evaluations, is the proposed solution reasonable and reliable? Moreover, suppose the statement is true, it lacks sufficient experiments to show that the order can be improved by the proposed solution.
- The alternative searching [22] and bias correction [16] methods are not first proposed in this paper. The authors should cite them in the mentioned positions in case readers misunderstand they are contributions of this paper.
- Section 3.3: Mixed-Precision Quantization: what is the difference between [9] and this paper? I do not seem novelty in this part.
- What is the significance of the mixed-precision setting of this paper (L236-237)? Most practical hardware platforms have supported int8 inference with good hardware performance.
- It lacks hyper-parameter analysis (e.g., alpha/beta/gamma/theta).
- Figure 1 and Equation (6,7) do not match. Figure 1 shows pre-layerNorm but Equation (6,7) uses post-layerNorm.

Updated:
I've read all the reviews and author responses. The authors addressed part of my concerns, e.g., explain the differences between cosine similarity and Pearson correlation and provide results of them. But some other major concerns are not addressed.

1) For ranking strategy, Table 3 actually shows better results over the baseline but the gain is very limited. More importantly, the idea of ranking is not well explained in the paper to demonstrate the order of attention values is crucial for quantizing transformer.
2) For nuclear norm, yes, this paper lacks necessary comparison experiments with the baseline of using Hessian Coefficient. Only stating that nuclear norm is easier than Hessian Coefficient is far insufficient to support this contribution.
3) For mix-precision quantization, I think more general or more challenging settings (e.g., using 2-8 mixed bits or using lower bits of {2,3,4} as HAWQ-v2 [9]) can be evaluated to make this part of paper stronger but they are missing. More importantly, the algorithm of mix-precision quantization is almost the same as [9] except the nuclear norm (but nuclear nom is not well compared with exiting work to support its contribution). So the technical novelty is limited.

In general, I think the main strength of the paper is that it brings best practices to realize post-training quantization to vision transformer, but the novelty is very limited: 1) low-bit quantization follows existing alternative searching [22] and bias correction [16] and 2) mix-precision quantization follows HAWQ-v2 [9], and the different algorithmic parts between this paper and previous methods are not well explained and validated. Thus, I think the quality of this paper does not reach the bar of NeurIPS and still vote for rejection.


**Time Spent Reviewing:**

4

---

> ### Author Response · Authors · 2021-08-10
> **Authors' Response**
>
> We would like to thank for the constructive and helpful comments.
>
> $\textbf{Q1:}$ The authors adopt Pearson correlation coefficient as the measurement for the similarity (L156-162). There are many other simpler choices for computing similarity (cosine, Euclidean, KL, etc). For example, the work of [22] uses cosine similarity [22], and the previous work of "Improving Post Training Neural Quantization: Layer-wise Calibration and Integer Programming" uses Euclidean distance. Why do the authors choose the Pearson correlation coefficient, and what are the advantages of this metric?
>
> $\textbf{A1:}$ Thanks for the nice question. The Person correlation coefficient for conducting the proposed method is identical to the normalized cosine similarity. The formulation can be presented as:
>
> $Cosine(x,y)=\frac{\sum_i x_i y_i}{\sqrt{\sum_i x_i^2}\sqrt{\sum_i y_i^2}}$
>
> $Person(x,y)=\frac{\sum_i (x_i-\overline{x}) (y_i-\overline{y}) } {\sqrt{\sum_i (x_i-\overline{x})^2} \sqrt{\sum_i (y_i-\overline{y})^2}} = Cosine(x-\overline{x},y-\overline{y})$
>
> It can be seen that Cosine similarity is not invariant to shifts so the Person correlation coefficient is more precise for evaluating the similarity since the mean value can be corrected by the bias correction. The experimental results in the following table demonstrate the analysis and we will add the analysis and the experimental results in the manuscript.
>
> | 8-bit ViT-B  | Euclidean distance | Cosine similarity | Pearson correlation |
> | :--: | :--: | :--: |  :--: |
> | Top-1 Accuracy | 75.42% | 75.57% | 75.81% |
>
> $\textbf{Q2:}$ Regarding ranking-aware quantization for self-attention, the authors state that "we empirically find that the relative order of the attention map has been changed after quantization as shown in Fig 1, which could cause a significant performance degradation" (L166-167). This "observation" is crucial to the main contribution and novelty of this paper, but it lacks theoretical analysis and sufficient experimental evaluations to demonstrate that the change of order actually negatively affects the final quantization accuracy. Without such analysis or evaluations, is the proposed solution reasonable and reliable? Moreover, suppose the statement is true, it lacks sufficient experiments to show that the order can be improved by the proposed solution.
>
> $\textbf{A2:}$ Thanks for the constructive quesiton. The difference of the rank before and after quantization is shown in the figure of the link (https://imgtu.com/i/ftNMkQ). And the difference can be compensated by introducing the ranking-loss as shown in the figure. The performance of the quantized model demonstrates the effectiveness of the proposed method.
>
> $\textbf{Q3:}$ The alternative searching [22] and bias correction [16] methods are not first proposed in this paper. The authors should cite them in the mentioned positions in case readers misunderstand they are contributions of this paper.
>
> $\textbf{A3:}$ Thanks for the suggestion and we will cite the paper in the mentioned positions.
>
> $\textbf{Q4:}$ Section 3.3: Mixed-Precision Quantization: what is the difference between [9] and this paper? I do not seem novelty in this part.
>
> $\textbf{A4:}$ We utilize weight nuclear norm to determine the bit-width of each layer in the paper and it is much easier compared to the Hessian Coefficient used in [9]. The values of weight nuclear norm in Figure.1 shows that they are various for different layers and the experiments demonstrate the effectiveness.
>
> $\textbf{Q5:}$ What is the significance of the mixed-precision setting of this paper (L236-237)? Most practical hardware platforms have supported int8 inference with good hardware performance.
>
> $\textbf{A5:}$ Thanks for the constructive comments. Although many platforms support int8 inference with good hardware performance, there are some platforms which can support mixed-precision as shown in [1]. Moreover, the performance of proposed 8-bit quantized model is also better than the previous methods.
>
> [1] HAQ: Hardware-Aware Automated Quantization with Mixed Precision.
>
> $\textbf{Q6:}$ It lacks hyper-parameter analysis (e.g., alpha/beta/gamma/theta).
>
> $\textbf{A6:}$ Thanks for the suggestion. The $\alpha$ and $\beta$ are employed to determine the quantization range of the weights or activations and they are set empirically. Besides, we have done some ablation study of $\gamma$ and $\theta$ and the results are shown in the following table. Note that the experiments are conducted with 8-bit ViT-B model on ImageNet dataset.
>
> | $\gamma$  | 0.05 | 0.1 | 0.2 | 0.3 |
> | :--: | :--: | :--: |  :--: | :-:|
> | Top-1 Accuracy | 75.63% | 75.81% | 75.42% | 74.09% |
>
> | $\theta$ | 0.1 | 0.2 | 0.3 | 0.5 |
> | :--: | :--: | :--: |  :--: | :-:|
> | Top-1 Accuracy | 75.77% | 75.81% | 75.39% | 74.18% |
>
> $\textbf{Q7:}$ Figure 1 and Equation (6,7) do not match. Figure 1 shows pre-layerNorm but Equation (6,7) uses post-layerNorm.
>
> $\textbf{A7:}$ Sorry for the typo and we will utilize the pre-LayerNorm for Eq.(6,7) in the manuscript.

---

### Public Comment · ~Natalia_Frumkin1 · 2022-08-26
**Code Unavailable**

This paper has a gitee link, however it gives a 404 error. I have emailed the authors, and have not heard a response. Going through the gitee repo, it seems that this code was never open-sourced.

---

> ### Public Comment · Authors · 2022-08-27
> **Code will be released soon**
>
> Sorry for the inconvenience. The code is not available due to the company regulation and some transfer problems. We have solved the problem recently and the code will be released soon.

---

> ### Public Comment · ~Foroozan_Karimzadeh1 · 2023-07-21
> **Code release**
>
> Hello,
>
> This is an interesting paper. Do you have any estimate when the code will be released? Thanks.

---

### Decision · Program_Chairs · 2021-09-28

**Decision:**

Accept (Poster)

**Comment:**

The manuscript has been reviewed by four experienced reviewers, among whom three voted for acceptance and one voted for rejection. There are several rounds of discussion post the release of the reviews, and the rejection reviewer (NVLj) raises some concerns, including the quality of 1) Fig.1, 2) comparisons to Hessian Coefficient, and 3) limited novelty. All three concerns are highly constructive, yet given other contributions of the submission acknowledged by other reviewers, the pros indeed overweight the cons. Specifically, despite reviewer NVLj believes that the submission lacks novelty, all the other three reviewers, in fact, viewed the proposed approach as being novel.

The AC agrees with the major vote and hence recommends an acceptance. However, it is strongly recommended that the authors also incorporate the comments from reviewer NVLj, especially on comparisons to Hessian Coefficient, into the final version, if possible.

**Consistency Experiment:**

NeurIPS has a long history of experimentation. In 2014, NeurIPS ran an experiment in which 10% of submissions were reviewed by two independent committees to quantify the randomness in the review process. This year, we repeated a variant of this experiment to see how the quality of the review process has changed over time.  This paper was part of the experiment and was therefore assigned to two committees (consisting of reviewers, an Area Chair, and a Senior Area Chair) that reached independent decisions.  If both committees made the same recommendation, this recommendation was followed. If a single committee recommended acceptance, the paper was accepted (with the exception of a few cases in which the other committee identified what we considered a fatal flaw, e.g., an error in a key result).

This copy’s committee reached the following decision: **Accept (Poster)**

The other committee assigned to the paper recommended **Reject**.  You can find the other set of reviews, along with any follow up discussion with the authors here:
https://openreview.net/forum?id=wCrH0JBCFNm